

# Associations between squalene epoxidase gene polymorphisms and obesity

Jia-Qing Yu, Feng-Xia Wang, Shuai Liu, Bing Zhu and Yi-Tong Ma

The First Affiliated Hospital of Xinjiang Medical University, Wulumuqi, Xinjiang, China

## ABSTRACT

**Background**. Among the known control points of cholesterol synthesis, squalene epoxidase (SQLE) is considered a key factor influencing cholesterol metabolism.

**Methods**. A total of 1,045 consecutive participants were divided into an obese group and a control group. Blood biochemical markers were measured, and deoxyribonucleic acid (DNA) was extracted from all participants. Statistical analyses were conducted to assess the associations between SQLE gene single nucleotide polymorphisms (SNPs) and obesity.

**Results**. The C/C genotype of SQLE SNP1 (rs10104486) was significantly more associated with obesity compared to the A/A genotype. A significant difference in genotype distribution frequency for rs10104486 was observed between the obese and control groups. The recessive model (CC *vs*. AC + AA) also showed a statistically significant difference. For SQLE SNP2 (rs2288312), differences were found in genotype distribution frequency, allele frequency, and the recessive model (GG *vs*. AA + AG) between the two groups.

**Conclusions**. This study indicates a correlation between the SQLE gene polymorphisms rs10104486 and rs2288312 and obesity in a young population. Participants carrying the C allele of rs10104486 were more likely to develop obesity than those carrying the A allele, with the CC genotype identified as a predisposing factor.

## INTRODUCTION

Studies have revealed a causal relationship between low-density lipoprotein cholesterol (LDL-C) levels and obesity. An increased risk of coronary atherosclerotic heart disease (CAD) is considered associated with abnormally high LDL-C levels in obese people (*Varbo et al., 2015*). Dyslipidaemia is the result of the interaction of both environmental and genetic factors (*Wijers, Kuivenhoven & Van De Sluis, 2015*). Abnormal cholesterol metabolism is also associated with a variety of human diseases (*NCD Risk Factor Collaboration, 2017*; *Bouchard, 2021*; *Zhao et al., 2022*). Various studies have shown that alterations in squalene epoxgenase (SQLE), a key enzyme involved in cholesterol biosynthesis, in the cholesterol synthesis pathway are associated with various diseases associated with in cholesterol metabolism (*Locke et al., 2015*; *Haeusler, McGraw & Accili, 2018*; *Speliotes et al., 2010*). A research has identified a correlation between the polymorphism at the rs9370867 locus of the IDOL gene and obesity. Compared to the control group, individuals carrying the

Corresponding author
Yi-Tong Ma, myt-xj@163.com

AG genotype exhibit a 1.482-fold increased risk of developing obesity. Similarly, those with the GG genotype at the rs2072783 locus show a 1.849-fold heightened risk of obesity. The presence of the G allele at both the rs9370867 and rs2072783 loci in the IDOL gene is associated with a greater likelihood of obesity compared to individuals carrying the A allele. The AG genotype at the rs9370867 locus and the GG genotype at the rs2072783 locus are identified as susceptibility factors for obesity; however, the underlying mechanisms remain unclear (*Liu et al., 2021*). However, genetic studies on the relationships between SQLE gene polymorphisms and obesity and blood lipid and metabolite levels are relatively rare. This study aimed to investigate the associations between lipid and metabolite levels and body mass index (BMI) and SQLE gene polymorphisms in a healthy young population.

## DATA AND METHODS

### Selection and grouping of the study participants

This study was a cross-sectional survey. We measured height and weight *via* the standard method (Seca, Seca213) and an electronic weight scale (Seca, Seca877), which are accurate to 0.1 cm and 0.1 kg, respectively, and we calculated BMI as weight (kg)/height$^2$ (m$^2$). We double-numbered the data according to the physical examination date, from earlier to later dates, to establish a database. We used an extreme phenotype research strategy in this study to determine the LDL-C levels of people of different sexes from lower to higher levels and ultimately included 1,045 participants after screening according to the inclusion and exclusion criteria. A total of 7,095 healthy individuals who underwent physical examination at the Health Examination Center of Xinjiang Medical University from October to November 2018 were included in this study. In accordance with the Asia–Pacific population criteria, individuals with a physical fitness index >25 kg/m$^2$ were included in the obese group ($n = 507$), whereas those with a BMI 18.5~22.99 kg/m$^2$ were included in the control group ($n = 538$) if they met the strict inclusion and exclusion criteria. This study was approved by the Ethics Committee of the First Affiliated Hospital of Xinjiang Medical University (approval number: 20211015-18), and all the participants provided written informed consent.

### Inclusion and exclusion criteria

The inclusion criteria were as follows: (1) no respiratory system, digestive system or endocrine system disease; (2) no history of liver disease, kidney disease, or cardiovascular or cerebrovascular system disease; and (3) no history of infectious disease or other related history of surgery; (4) no history of smoking or drinking. The participant exclusion criteria were as follows: (1) no destruction of blood samples (such as through transportation, cold storage, extraction failure, or coagulation); (2) samples sent for DNA extraction were lost or no DNA was detected during the sequencing process; and (3) basic information and physical examination data were missing from the general database.

### Blood biochemical marker measurement and DNA extraction

All study participants were required to fast and drink only water for 8–10 h before the morning of blood collection. A five mL blood sample was drawn from the cubital vein. Four

millilitres of this sample was used to measure total bilirubin (TBIL), blood urea nitrogen (BUN), creatinine (Cr), alanine aminotransferase (ALT), aminotransferase (AST), total cholesterol (TC), fasting blood glucose (FBG), triglyceride (TG), high-density lipoprotein cholesterol (HDL-C) and LDL-C levels. The remaining one mL blood sample was used for DNA extraction, and the eligible DNA samples were genotyped *via* the SNPscan technique. To ensure accuracy, we received specialized technical training at Tianhao Biological Company. During the extraction of DNA, blood samples were promptly transported to the Key Laboratory of Cardiovascular Disease Research at the First Affiliated Hospital of Xinjiang Medical University, where they were sealed in anticoagulant-treated collection tubes and stored at $-80$ °C until DNA extraction. Prepared isopropanol and 70% ethanol (350 mL anhydrous ethanol combined with 150 mL distilled water). To achieve optimal experimental results, blood samples that have undergone more than three freeze-thaw cycles should not be used, as this may increase the risk of contamination and reduce the quantity of DNA extraction. The detailed experimental procedure is as follows: (1) Take one mL of EDTA-anticoagulated peripheral blood, incubate at 37 °C after thawing, then transfer to a 50 mL centrifuge tube. Add seven mL of cell lysis buffer, vortex vigorously to ensure thorough mixing, and centrifuge at 4,000 rpm at low temperature for 10–15 min. Discard the supernatant, observing a pale red precipitate at the bottom of the tube; (2) Subsequently, add five ml of cell lysis buffer to a 50 ml centrifuge tube, invert and mix thoroughly, and let it stand for 10–15 min. Centrifuge at 4,000 rpm for 5 min, then carefully discard the supernatant and allow to drain excess liquid; (3) Add two ml of buffer solution FG to the test tube, followed by the addition of the protease K mixture, and vortex rapidly to ensure thorough mixing; (4) The test tube is placed in a water bath at 56 °C for 60 min or in an oven for approximately 1 h; (5) Remove the test tube from the water bath and transfer the pale yellow liquid into two two mL centrifuge tubes; (6) Add two mL of isopropanol to the test tube, then invert vigorously to mix thoroughly until a filamentous or clustered white precipitate is formed; (7) Isolate the white precipitate and rinse it 2–3 times with 70% ethanol, then allow it to air dry; (8) Add 400–500 µL of TE buffer and incubate at room temperature or 4 °C for over 10 h to ensure complete dissolution of the DNA. Subsequently, extract two µL of the solution for concentration measurement and purity assessment using quantitative analysis methods (Nanodrop, Wilmington, DE, USA; agarose gel electrophoresis, and Qubit, Paris, France). After the quality evaluation, store the samples at $-80$ °C for future use. Subsequently, extract two µL of the solution for concentration measurement and purity assessment using quantitative analysis methods (Nanodrop, Wilmington, DE, USA; agarose gel electrophoresis, and Qubit, Paris, France). After the quality evaluation, store the samples at $-80$ °C for future use. Simultaneously, precise operations are conducted in both DNA identification and quantification. The specific operations are described as follows: (1) Genomic DNA dissolved in distilled water or TE (pH 8.0); (2) Sample requirement: for Qubit detection, a concentration of at least 20 ng/µl is required, with a total amount of at least 1.5 µg; (3) Sample Integrity: the integrity of genomic DNA is assessed by agarose gel electrophoresis, which requires that the electrophoretic bands are clearly visible, without significant degradation, and free from RNA contamination; (4) Sample purity: Nanodrop detection, OD260/280 =

1.8~2.2, OD260/230 $\geq$ 2.0; (5) Transportation: DNA is transported at low temperatures (with the addition of dry ice at $-70\,°C$), with each centrifuge tube sealed using Parafilm. Single-nucleotide polymorphism (SNP) site alleles were identified on the basis of the high specificity of the ligase chain reaction. We introduced different lengths of nonspecific sequences at the end of the ligation probe. Similarly, a ligase chain reaction was used to obtain ligation products of different lengths corresponding to the sites, and then, polymerase chain reaction (PCR) was used to amplify the ligation products that were fluorescently labelled with universal primers. We then separated the amplified products *via* fluorescent capillary electrophoresis and used GeneMapper software to obtain the genotypes at the SNP sites.

### Selection of SNP loci and detection of genetic variants

We downloaded the relevant gene sequences from the HapMap database and detected gene polymorphism loci *via* Haploview 4.2 software (the parameter conditions were set as $r^2 > 0.8$ and minimum allele frequency (MAF) $\geq 0.05$). In our previous SNP studies, we found that a single SNP may not necessarily reveal an association with the disease, but two or more SNPs may have a combined effect. To ensure meaningful results, we decided to select two SNPs. We reviewed the relevant reports on the NCBI website and selected the rs10104486 (SNP 1) and rs2288312 (SNP 2) gene loci. We found that the two genetic loci rs10104486 (SNP1) and rs2288312 (SNP2) of SQLE have rarely been reported, so we selected these two loci and performed genotype typing. We used SNPscan typing technology to genotype the above SNP sites, and we conducted quality control inspections through the analysis of double-blind samples and negative controls for Hardy–Weinberg equilibrium (HWE) and MAF.

### Statistical analysis

SPSS 26.0 statistical software (IBM Corp., Armonk, NY, USA) was used for the statistical processing of the data. The measurement data are expressed as the means $\pm$ standard deviations ($\bar{x} \pm s$). Means were compared between two groups *via* the $t$ test. One-way analysis of variance (ANOVA) was used for multigroup comparisons, and counting data were compared *via* the chi-square test. We performed the HWE test on the number and frequency of genotypes in two SNPs in the SQLE gene (rs10104486 and rs2288312) to confirm the population representativeness of the samples, and we compared the distribution frequencies between different genotypes *via* the chi-square test. Unconditional logistic regression was used to evaluate the correlation between the SQLE gene SNP and obesity (test level $\alpha = 0.05$). $P < 0.05$ was considered to indicate a statistically significant difference.

## RESULTS

### Comparison of the general data between the obesity group and the control group

This study included 507 obese participants (318 males and 189 females) and 538 control participants (259 males and 279 females). Compared with those in the control group,

**Table 1  Comparison of general data between the obese and control groups.**

| Index | Control group (n = 538) | Obesity group (n = 507) | $t//X^2$ | P |
|---|---|---|---|---|
| Sex (n (%)) | | | | |
| Male | 259 (48.1) | 318 (62.7) | 22.4 | <0.001 |
| Female | 279 (51.9) | 189 (37.3) | | |
| Age (years) | 20.77 ± 1.49 | 20.86 ± 1.51 | −0.973 | 0.331 |
| Systolic blood pressure (mmHg) | 108.64 ± 10.96 | 117.05 ± 10.85 | −12.449 | <0.001 |
| Diastolic blood pressure (mmHg) | 67.71 ± 7.17 | 73.74 ± 9.42 | −11.687 | <0.001 |
| HDL-C (mmol/L) | 1.42 ± 0.28 | 1.24 ± 0.24 | 11.350 | <0.001 |
| LDL-C (mmol/L) | 2.18 ± 0.51 | 2.45 ± 0.6 | −7.991 | <0.001 |
| TGs (mmol/L) | 0.84 ± 0.35 | 1.16 ± 0.71 | −9.075 | <0.001 |
| Cr (μmol/L) | 76.45 ± 11.26 | 80.39 ± 15.12 | −4.799 | <0.001 |
| TC (mmol/L) | 3.9 ± 0.62 | 4.08 ± 0.74 | −4.158 | <0.001 |
| FPG (mmol/L) | 5 ± 0.38 | 5.06 ± 0.59 | −1.753 | 0.084 |
| BUN (mmol/L) | 4.43 ± 1.15 | 4.46 ± 1.1 | −.315 | 0.753 |
| AST (U/L) | 19.41 ± 7.73 | 21.04 ± 16.8 | −2.026 | <0.001 |
| ALT (U/L) | 18.84 ± 14.85 | 24.72 ± 18.14 | −5.745 | 0.047 |
| TBIL (μmol/L) | 13.92 ± 6.21 | 13.54 ± 5.64 | 1.054 | 0.292 |

**Table 2  Hard–Weinberg equilibrium test results.**

| Genotype | $X^2$ | P |
|---|---|---|
| rs10104486 | 0.32 | 0.85 |
| rs2288312 | 0.21 | 0.9 |

HDL-C, FBG, TG, and LDL-C levels were significantly greater in the obese group than in the control group, as were the incidence of hypertension and history of alcohol use ($P < 0.05$). There were no significant differences in diabetes incidence, smoking history or cholesterol levels between the two groups ($P > 0.05$), as shown in Table 1.

### Distribution of HWE test results and genotype and allele frequencies

The genotype frequencies of two tag SNP sites (rs10104486 and rs2288312) in the SQLE gene were consistent with HWE ($P > 0.05$) in both groups and were population representative, as shown in Table 2. The frequency of SNP 1 (rs10104486) was different between the obese patients and controls, as was the frequency of the genotypes in the recessive model (CC vs. AC + AA). The differences in the distributions of genotype frequency and allele frequency and in the frequency of the genotypes in the recessive model (GG vs. AA + AG) were statistically significant ($P < 0.05$) for SNP 2 (rs2288312), as shown in Table 3.

### Association between the rs10104486 genotype polymorphism and obesity

After controlling for age, sex and other factors, we found that the likelihood of obesity was significantly greater in participants with the C/C genotype of rs10104486 than in those with the A/A genotype (Odds ratio (OR) = 1.545, 95% confidence interval (CI) [1.042–2.291],

**Table 3 Comparison of the distribution of the genotype and allele frequencies in each model in the obese and control groups ($n$(%)).**

| SNPs | Genotype | Control group | Obesity group | $X^2$ | $P$ |
|---|---|---|---|---|---|
| SNP1 | AA | 238 (44.24) | 203 (40.04) | 8.3 | 0.016 |
| (rs10104486) | AC | 244 (45.35) | 221 (43.59) | | |
| | CC | 56 (10.41) | 83 (16.37) | | |
| Explicit model | AA | 238 (44.24) | 203 (40.04) | 1.9 | 0.17 |
| | AC+CC | 300 (55.76) | 304 (59.96) | | |
| Additive model | AC | 244 (45.35) | 221 (43.59) | 0.329 | 0.57 |
| | AA+CC | 294 (54.65) | 286 (56.41) | | |
| Recessive model | CC | 56 (10.41) | 83 (16.37) | 8.045 | 0.005 |
| | AA+AC | 482 (89.59) | 424 (83.63) | | |
| Allelic genes | A | 720 (66.91) | 627 (61.83) | 5.9 | 0.15 |
| | C | 356 (33.09) | 387 (38.17) | | |
| SNP2 | AA | 236 (43.87) | 204 (40.24) | 6.32 | 0.042 |
| (rs2288312) | AG | 237 (44.05) | 214 (42.21) | | |
| | GG | 65 (12.08) | 89 (17.55) | | |
| Explicit model | AA | 236 (43.87) | 204 (40.24) | 1.411 | 0.2 |
| | AG+GG | 302 (56.13) | 303 (59.76) | | |
| Additive model | AG | 237 (44.05) | 214 (42.21) | 0.36 | 0.5 |
| | AA+GG | 301 (55.95) | 293 (57.79) | | |
| Recessive model | GG | 65 (12.08) | 89 (17.55) | 6.2 | 0.01 |
| | AA+AG | 473 (87.92) | 418 (82.45) | | |
| Allelic genes | A | 709 (65.89) | 622 (61.34) | 4.7 | 0.03 |
| | G | 367 (34.11) | 392 (38.66) | | |

**Table 4 Effects of the rs10104486 and rs2288312 genotype polymorphisms on the incidence of obesity.**

| SNP | Group | Genotype | | |
|---|---|---|---|---|
| | | A/A | A/C | C/C |
| rs10104486 | | | | |
| | Control group | 238 | 44 | 56 |
| | Obesity group | 203 | 221 | 83 |
| | OR (95% CI) | 1 | 1.127 (0.864~1.471) | 1.545 (1.042~2.291) |
| | P | | 0.377 | 0.031 |

$P < 0.05$). The above results revealed that the C/C genotype can act as an independent risk factor for obesity, as shown in Table 4.

## Logistic regression analysis of SQLE gene SNP 1 (rs10104486) and SNP 2 (rs2288312) in the obesity and control groups

Unconditional logistic regression was used to correct for age, HDL-C levels, sex, ethnicity, LDL-C levels, and TG levels, and the recessive model of rs10104486 (CC *vs.* AA + AC) and the stealth model of rs2288312 (GG *vs.* AA + AG) did not constitute obvious risk factors for obesity, as shown in Table 5.

**Table 5  Logistic regression analysis of SQLE gene SNP 1 (rs10104486) and SNP 2 (rs2288312) in the obesity and control groups.**

| Index | B | SE | Wald | OR (95% CI) | P |
|---|---|---|---|---|---|
| SNP1 | | | | | |
| Recessive model (CC *vs.* AA+AC) | −0.217 | 0.218 | 0.995 | 0.81(0.53∼1.23) | 0.319 |
| Age/year | −0.037 | 0.049 | 0.561 | 0.96 (0.88∼1.06) | 0.454 |
| Sex | −0.094 | 0.201 | 0.22 | 0.91 (0.61∼1.35) | 0.639 |
| Nationality | 0.423 | 0.151 | 7.815 | 1.53 (1.14∼2.05) | 0.005 |
| Blood pressure | −24.381 | 4,248.842 | 0 | – | 0.995 |
| HDL-C (mmol/L) | 4.078 | 0.924 | 19.454 | 59 (9.64∼361.22) | <0.001 |
| LDL-C (mmol/L) | 1.503 | 0.92 | 2.668 | 4.5 (0.74∼27.29) | 0.102 |
| TGs (mmol/L) | −0.195 | 0.23 | 0.719 | 0.82 (0.52∼1.29) | 0.397 |
| Cr (μmol/L) | −0.019 | 0.007 | 6.996 | 0.98 (0.97∼1) | 0.008 |
| TC (mmol/L) | −1.885 | 0.833 | 5.121 | 0.15 (0.03∼0.78) | 0.024 |
| FPG (mmol/L) | 0.044 | 0.156 | 0.079 | 1.05 (0.77∼1.42) | 0.778 |
| BUN (mmol/L) | 0.126 | 0.067 | 3.597 | 1.14 (1∼1.29) | 0.058 |
| AST (U/L) | −0.025 | 0.009 | 8.007 | 0.98 (0.96∼0.99) | 0.005 |
| ALT (U/L) | 0.029 | 0.015 | 3.931 | 1.03 (1∼1.06) | 0.047 |
| TBIL (μmol/L) | 0.019 | 0.013 | 2.238 | 1.02 (0.99∼1.05) | 0.135 |
| SNP2 | | | | | |
| Recessive model (GG *vs.* AA+AG) | −0.206 | 0.208 | 0.979 | 0.81 (0.54∼1.22) | 0.322 |
| Age/year | −0.036 | 0.049 | 0.548 | 0.96 (0.88∼1.06) | 0.459 |
| Sex | −0.093 | 0.201 | 0.213 | 0.91 (0.61∼1.35) | 0.644 |
| Nationality | 0.427 | 0.151 | 8.014 | 1.53 (1.14∼2.06) | 0.005 |
| Blood pressure | −24.55 | 4,224.004 | 0 | – | 0.995 |
| HDL-C (mmol/L) | 4.051 | 0.924 | 19.207 | 57.43 (9.39∼351.46) | <0.001 |
| LDL-C (mmol/L) | 1.473 | 0.92 | 2.56 | 4.36 (0.72∼26.49) | 0.11 |
| TG (mmol/L) | −0.2 | 0.23 | 0.757 | 0.82 (0.52∼1.29) | 0.384 |
| Cr (μmol/L) | −0.019 | 0.007 | 7.067 | 0.98 (0.97∼1) | 0.008 |
| TC (mmol/L) | −1.856 | 0.833 | 4.964 | 0.16 (0.03∼0.8) | 0.026 |
| FPG (mmol/L) | 0.052 | 0.156 | 0.112 | 1.05 (0.78∼1.43) | 0.738 |
| BUN (mmol/L) | 0.128 | 0.067 | 3.663 | 1.14 (1∼1.3) | 0.056 |
| AST (U/L) | −0.025 | 0.009 | 8.151 | 0.98 (0.96∼0.99) | 0.004 |
| ALT (U/L) | 0.03 | 0.015 | 4.117 | 1.03 (1∼1.06) | 0.042 |
| TBIL (μmol/L) | 0.019 | 0.013 | 2.29 | 1.02 (0.99∼1.05) | 0.13 |

## DISCUSSION

The number of people with cardiovascular diseases is increasing every year. Recently, several studies have shown that obesity and an excessive increase in LDL-C are the two main risk factors for cardiovascular disease (CVD) and that there is a causal relationship between CVD and these factors (*Varbo et al., 2015*). Obesity is a metabolic syndrome that disrupts the metabolic balance of the human body. Fat accumulation occurs when the body consumes too much energy. Obesity is now considered a global epidemic that can increase

the risk of a variety of diseases. In 2016, according to the World Health Organization (WHO), more than 1.9 billion adults were overweight (39% of the population), whereas more than 650 million people (13% of the population) were obese (*Mandel et al., 2010*). Obesity can significantly induce various cardiovascular diseases. Previous studies have shown that environmental and genetic factors simultaneously influence the development of obesity and that BMI in the overall population is influenced by environmental factors, whereas the BMI distribution of individuals is mainly determined by genetic factors (*Natarajan et al., 2018*).

According to the report in the Guidelines for the Prevention and Treatment of Dyslipidaemia in China (2016 Revision), the prevalence of hyperlipidaemia has gradually increased in recent years (*Yuan et al., 2023*). Dyslipidaemia increasingly affects the development of atherosclerotic CVD. Dyslipidaemia is characterized by increased TC or LDL-C levels. Both elevated LDL-C and elevated TC are causally associated with the emergence of obesity, and one of the most important reasons for the increased risk of CAD in obese individuals is the increase (*Varbo et al., 2015*) in LDL-C. Low TC and LDL-C levels are associated with a low incidence of atherosclerotic cardiovascular disease (*Dos Santos et al., 2022*). When dyslipidaemia occurs, doctors usually recommend that individuals change their habits (such as quitting smoking, limiting alcohol use, exercising more and eating a low-fat diet) to reduce cholesterol or TGs. Approximately 70% of LDL-C in human blood enters all tissue cells through the LDLR-mediated classic endocytosis pathway, and the remainder is ingested *via* cell-mediated pathways, such as scavenger receptor and nonreceptor pathways (*Xu et al., 2022*).

In recent years, additional research has focused on obesity, dyslipidaemia and genetic factors. Genetic polymorphisms are considered important factors of cardiovascular disease (such as early-onset coronary heart disease) (*Li et al., 2023*). Previous studies have shown that leptin is an anorexigenic hormone produced during adipose tissue formation. Leptin can cross the blood–brain barrier through a specific transport pathway and thereby convey information on the lipid content in adipose tissue to the hypothalamus, thus regulating lipid metabolism. The MC4R gene is involved in regulating the leptin signalling pathway. Patients with MC4R frameshift mutations develop significant SIM1 deletions that cause reduced MC4R synthesis, and these mutations also cause obesity (*Guo et al., 2023*). The insulin signalling pathway has also been shown to be important for the regulation of energy metabolism balance. When blood glucose increases in the body, insulin secretion is activated. Several studies have shown that alterations in genes related to insulin signalling, such as HHEX-IDE, KCNQ1, MNTR1B, and GIPR, cause changes in BMI. Furthermore, AMY 1 gene mutation can affect the concentration of salivary amylase, subsequently affecting the sweetness of food and human appetite (*Rask-Andersen et al., 2019*).

Recent studies have shown that sophisticated approaches have been developed to regulate the transcriptional and posttranslational levels of cholesterol (*Howe et al., 2017*). The SQLE protein is found mainly in the endoplasmic reticulum. This protein is encoded by the human SQLE gene and is an important downstream cholesterol synthesis rate-limiting enzyme (*Gudmundsson et al., 2007*; *Gill et al., 2011*) located on the long arm 24.1 of human chromosome 8. Metabolic gene expression analysis clearly indicated that the reduction

in and loss of SQLE expression are the causes of cellular cholesterol auxotrophy. The biological function of SQLE is to convert squalene to 2–3-oxidized squalene, which is further involved in the synthesis of sterol and cholesterol *in vivo*, as well as the regulation of cellular metabolism and organ specificity. Loss of this enzyme in the cholesterol synthesis pathway leads to the accumulation of the upstream metabolite squalene, which then changes the cytoplasmic profile to provide a growth advantage under oxidative stress conditions (*Ma et al., 2021*). Several studies have shown that the protein expression of glucose transporter 1 (GLUT1) and lactate dehydrogenase (LDH) decreases significantly after SQLE expression is knocked down. Cell glucose consumption levels, lactate levels and adenosine triphosphate (ATP) levels were significantly reduced after the expression of SQLE was inhibited. Some scholars have speculated that SQLE may regulate the AKT/mTOR signalling pathway and affect glucose metabolism, while lipid metabolism is also closely associated with glucose metabolism and causes changes in BMI (*Speliotes et al., 2010*). At present, studies on the role of SQLE gene polymorphisms in cholesterol metabolism and the effects of these polymorphisms on obesity are in the primary stage, and identifying the specific mechanism of action of SQLE gene polymorphisms is important.

The liver is an important organ for glucose metabolism and cholesterol synthesis and metabolism and is closely related to the regulation of blood sugar and blood lipids. Both ALT and AST originate from the cytoplasm of hepatocytes. Studies have shown that elevated AST levels are associated with elevated blood glucose levels, insulin resistance and metabolic syndrome (MS), including cardiovascular disease and atherosclerosis, and that the rate of ALT abnormalities decreases with age. ALT can perform its normal physiological function in a healthy state. When hepatocytes are damaged for various reasons, ALT is released into the blood, causing abnormal ALT levels in the blood. Studies have shown that obesity, age, and HDL-C levels are factors that influence ALT (*Wang & Guo, 2019*), whereas obesity is associated with abnormal lipid metabolism (*Zhang & Huang, 2020*). Approximately 50% of LDL-C is cholesterol, and after further research, some scholars previously reported that LDL-C was positively related to ALT. These authors speculated that increased lipids (represented by TGs and TC), decreased lipid clearance ability (represented by HDL-C), and fatty liver impairment (represented by ALT) were causally related to increased serum LDL-C in patients with fatty liver.

In summary, this study reported the association between SQLE gene polymorphisms and obesity in a young population. This study found statistically significant differences in the levels of ALT, AST, TC, TG, LDL-C, HDL-C, and CREA between the obese group and the control group. The results of this study indicate a correlation between the rs10104486 and rs2288312 SQLE gene polymorphisms and obesity in a young population. Young participants carrying the C allele of the SQLE gene rs10104486 polymorphism were more likely to develop obesity than those carrying the A allele, and the CC genotype was a predisposing factor for obesity. These findings could lead to new insights at the biomolecular and genetic levels for key interventions for the prevention and treatment of fatty liver disease and cardiovascular disease in young obese people and for improving blood lipid metabolism to avoid early liver and coronary artery damage. Whether the dominant or recessive model can constitute protective or risk factors has not been determined, and

the relevant mechanism has not been fully defined. In this study, we focused solely on the association between gene polymorphisms and body mass index (BMI), which does not comprehensively elucidate the mechanisms and functions of these two SNPs. This represents a limitation of the present research. Therefore, we anticipate that future research will involve additional cellular and animal experiments to further elucidate the molecular mechanisms and functional roles of these two SNPs. In this study, we did not employ multiple hypothesis testing corrections (such as Bonferroni, FDR), which may lead to a potential bias in the experimental results due to the first type error rate. Future research will clarify the relevant mechanism involved. Moreover, the sample size was relatively small, so further expanding the sample size is necessary, and the research methods should be further improved. We can study this topic in greater depth in the later stage. We will continue to conduct in-depth research on the aforementioned achievements, exploring the mechanisms and functions of genotypic biomolecules, as well as their association with obesity.

## ACKNOWLEDGEMENTS

The authors would like to thank AJE for the English language review.

### Funding

This work was funded by the special fund project for the Central Government-Guided Local Science and Technology Development (ZYYD2022A01). The funders had no role in study design, data collection and analysis, decision to publish, or preparation of the manuscript.

### Grant Disclosures

The following grant information was disclosed by the authors:
Central Government-Guided Local Science and Technology Development: ZYYD2022A01.

### Competing Interests

The authors declare there are no competing interests.

### Author Contributions

- Jia-Qing Yu conceived and designed the experiments, performed the experiments, analyzed the data, prepared figures and/or tables, authored or reviewed drafts of the article, and approved the final draft.
- Feng-Xia Wang conceived and designed the experiments, prepared figures and/or tables, and approved the final draft.
- Shuai Liu conceived and designed the experiments, authored or reviewed drafts of the article, and approved the final draft.
- Bing Zhu conceived and designed the experiments, authored or reviewed drafts of the article, and approved the final draft.

- Yi-Tong Ma conceived and designed the experiments, prepared figures and/or tables, and approved the final draft.

## Human Ethics

The following information was supplied relating to ethical approvals (i.e., approving body and any reference numbers):

This research was approved by the Ethics Committee of First Affiliated Hospital of Xinjiang Medical University (20211015-18).

## Data Availability

Raw data is available in the Supplemental Files.

## Supplemental Information

Supplemental information for this article can be found online at http://dx.doi.org/10.7717/peerj.19635#supplemental-information.

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
