# Peer review of "Associations between squalene epoxidase gene polymorphisms and obesity"

_PeerJ, doi:10.7717/peerj.19635_

## Round 0.1 · original submission · Minor Revisions

·

Basic reporting

Why did the authors choose the two SQLE SNPs rs10104486 and rs2288312? Was it based on functional annotations from previous studies, MAF (minimum allele frequency), or GWAS signals?

Are OR values ​​(odds ratios) and 95% confidence intervals reported? Is the observed effect size biologically meaningful?

Although the article points out that rs10104486 and rs2288312 are related to obesity, it does not discuss whether these variant sites are located in regulatory regions, affect SQLE expression or downstream cholesterol synthesis pathways, and whether there are functional experiments to support their biological mechanisms.

Experimental design

The article does not describe the technical methods used for DNA extraction and SNP typing (such as PCR-RFLP, TaqMan, sequencing, etc.). How accurate and reproducible are they?

Did the authors perform a Hardy-Weinberg equilibrium test on the SNP distribution of the control group? If not, the results of this test may affect the reliability of statistical inferences.

Validity of the findings

Did they adjust for potential confounding factors (such as diet, physical activity, family history, etc.)?

For multiple SNPs and multiple comparison models, were multiple hypothesis testing corrections (such as Bonferroni, FDR) performed to control the first type error rate?

Additional comments

What statistical models did the authors use to evaluate the association between genotype and obesity?

·

Basic reporting

the manuscript is clear, introduction part is poor in exact correlation between the studied parameters and their role in obesity.

role of genetic polymorphism in the involvement of obesity should focused.

tables are adequate in data.

Experimental design

the design is fair

Validity of the findings

the findings are valid but authors should focus on the findings as they are.

presenting both SNP effects on obesity, determining the role of each genotype.

statistically the results must represent in OR(C.I95%) to all the genotype comparisons

---

## Round 0.2 · accepted · Accept

Thank you for addressing the reviewer feedback and comments. Your article is now suitable for publication.

·

Basic reporting

no comment

Experimental design

no comment

Validity of the findings

no comment